# Upper extremity deep vein thrombosis in COVID-19: Incidence and correlated risk factors in a cohort of non-ICU patients

Nicola Mumoli[1]*, Francesco Dentali[2], Giulia Conte[1], Alessandra Colombo[1], Riccardo Capra[1], Cesare Porta[1], Giuseppe Rotiroti[1], Francesca Zuretti[3], Marco Cei[1], Flavio Tangianu[3], Isabella Evangelista[1], Josè Vitale[1], Antonino Mazzone[1], Igor Giarretta[3]

1 Department of Internal Medicine, ASST Ovest Milanese, Magenta (MI), Italy, 2 Department of Medicine and Surgery, Insubria University, Varese, Italy, 3 Department of Internal Medicine, ASST Sette Laghi, Varese, Italy

* nimumoli@tiscali.it

**Data Availability Statement:** All relevant data are within the manuscript and its Supporting Information files.

## Abstract

### Background

Venous thromboembolism is a frequent complication of COVID-19 infection. Less than 50% of pulmonary embolism (PE) is associated with the evidence of deep venous thrombosis (DVT) of the lower extremities. DVT may also occur in the venous system of the upper limbs especially if provoking conditions are present such as continuous positive airway pressure (CPAP). The aim of this study was to evaluate the incidence of UEDVT in patients affected by moderate-severe COVID-19 infection and to identify potential associated risk factors for its occurrence.

### Methods

We performed a retrospective analysis of all patients affected by moderate-severe COVID-19 infection admitted to our unit. In accordance with the local protocol, all patients had undergone a systematic screening for the diagnosis of UEDVT, by vein compression ultra-sonography (CUS). All the patients were receiving pharmacological thromboprophylaxis according to international guidelines recommendations. Univariate and multivariate analyses were used to identify risk factors associated with UEDVT.

### Results

257 patients were included in the study, 28 patients were affected by UEDVT with an incidence of 10.9% (95% CI, 7.1–14.7). At univariate analysis UEDVT appeared to be significantly associated (p< 0.05) with pneumonia, ARDS, PaO2/FiO2, D-dimer value higher than the age adjusted cut off value and need for CPAP ventilation. Multivariate analysis showed a significant association between UEDVT and the need for CPAP ventilation (OR 5.95; 95% IC 1.33–26.58). Increased mortality was found in patients affected by UEDVT compared to those who were not (OR 3.71; 95% CI, 1.41–9.78).

**Funding:** The author(s) received no specific funding for this work.

**Competing interests:** The authors have declared that no competing interests exist.

## Conclusions

UEDVT can occur in COVID-19 patients despite adequate prophylaxis especially in patients undergoing helmet CPAP ventilation. Further studies are needed to identify the correct strategy to prevent DVT in these patients.

## Introduction

Current evidence reports an increased risk of venous thromboembolism (VTE) in patients with Coronavirus Disease 2019 (Covid-19) admitted to both intensive care units (ICU) and non-intensive wards [1–5].

Several mechanisms may contribute to the procoagulant state in severe acute respiratory syndrome coronavirus 2 (SARS-CoV-2) infection. Firstly, it has been demonstrated that during COVID-19 there is an inflammatory state that causes endothelial cell dysfunction and leads to increased thrombin generation and impaired fibrinolysis [6]. Secondly, hypoxia can stimulate thrombosis by increasing blood viscosity and by inducing transcription factor-dependent signaling pathways [7]. Thirdly, a number of potential risk factors for VTE, including infection, immobilization, respiratory failure, and central venous catheter use, are common in these patients.

A high number of patients with moderate-severe COVID-19 disease develop acute respiratory distress syndrome (ARDS) and require respiratory support [8, 9]. In patients presenting with signs and symptoms of ARDS, several international guidelines recommend the use of continuous positive airway pressure (CPAP) [10] with the aim of avoiding intubation and reducing the number of patients requiring ICU admission [11]. CPAP delivers a constant flow of oxygen at a prescribed pressure which prevents alveolar collapse, improves gaseous exchange, and reduces respiratory fatigue. It has been suggested that early, rather than delayed, CPAP application to hypoxemic COVID-19 patients may lead to better outcomes and reduced mortality [12]. Collaterally, CPAP application causes venous stasis [13] and may be associated with an increased risk of VTE.

It is important to point out that, in patients affected by COVID-19 who are diagnosed with pulmonary embolism (PE), the presence of concomitant deep vein thrombosis (DVT) is detectable in less than 50% of cases [14]. A possible explanation for this mismatch is that, in almost all of the published studies, the presence of DVT was ruled out only in the lower extremities; however, the venous system of the upper limbs can also be involved by the thrombotic process especially if concomitant risk factors, such as venous catheters or venous stasis, are present.

The aim of our study was to evaluate the incidence of upper extremity deep venous thrombosis (UEDVT) in patients hospitalized for moderate-severe SARS-CoV-2 infection undergoing recommended antithrombotic prophylaxis and to identify potential risk factors associated with the occurrence of UEDVT.

## Methods

### Study population

Patients hospitalized between March 8th and April 28th and between October 2th and December 4th, 2020 for acute moderate-severe COVID-19 infection in the Covid-units of Magenta Hospital (Italy), were retrospectively evaluated for study inclusion. Subjects admitted to the

ICU were excluded from the study. Moderate disease was defined as evidence of lower respiratory tract disease during clinical evaluation or imaging and oxygen saturation (SpO2) ≥94% in ambient air at sea level. Severe COVID-19 infection was defined as SpO2 <94% in ambient air at sea level, a ratio of arterial partial pressure of oxygen to the fraction of inspired oxygen (PaO2 / FiO2) <300 mm Hg, respiratory rate > 30 breaths / min, or lung infiltrates > 50% [15]. Patients <18 years of age, patients on full dose antithrombotic therapy (low molecular weight heparin, unfractionated heparin, fondaparinux, vitamin K antagonists, or direct oral anticoagulants) at arrival to the Emergency Room (ER) and patients with any previous VTE were excluded from the study.

## Procedures

Diagnosis of COVID-19 infection was made according to World Health Organization interim guidance [16] and confirmed by reverse-transcriptase–polymerase-chain-reaction (rt-PCR) assays performed on nasopharyngeal swab specimens in the clinical laboratory of Legnano Hospital, Milan, Italy.

Patients' data were manually recorded from electronic health records with the use of a quality-control protocol and structured abstraction tool. Patient confidentiality was protected by assigning a de-identified patient identification and electronic data were stored in a locked, password-protected computer. Data included demographics, comorbid conditions, laboratory values and radiographic findings on presentation. Laboratory and radiographic testing were performed according to clinical care needs and analyzed/interpreted on-site. First laboratory values within 48 hours of arrival to the ER were recorded from the electronic health dossier. Laboratory values included complete blood count, creatinine, D-dimer, and C-reactive protein (CRP). Congenital or acquired thrombophilia was not systematically ruled out in all patients, however it was reported, if present, in the electronic health records.

All patients with COVID-19 admitted by our team received enoxaparin 4000 UI (40 mg) subcutaneously (SC) once daily (OD). Patients with a creatinine clearance under 30 mL/min, as estimated by using the Cockcroft-Gault formula ([(140-age) × body weight × 0.85 if female] / (72 × serum creatinine) x 1.73 $m^2$ of body surface area), received enoxaparin 2000 UI (20 mg) SC OD. According to the local COVID-19 patient management protocol, screening for upper and lower limbs vein thrombosis was performed in all patients. Patients were not routinely screened for PE, but computed-tomography pulmonary angiogram (CTPA) was performed based on clinical suspicion.

## Sonographic evaluation

B-mode compression ultrasound (CUS) imaging was performed with a high-resolution linear probe (ML6-15 MHz), using a General Electric Healthcare Vivid T8 ultrasound system, and consisted in the assessment of the superficial and deep venous systems of all four limbs. Settings for the ultrasound scanner were adjusted and standardized for all examinations. The deep veins included in the study were the internal jugular, brachiocephalic, subclavian, axillary, brachial veins for the upper limb and femoral, popliteal and distal veins (posterior tibial, fibular, gastrocnemius, and soleal veins) for the lower limb. The superficial veins examined were basilic vein, cephalic vein and distal superficial veins for the upper limb and great and small saphenous veins for the lower limb.

All venous segments were examined in real-time B-mode and with color-Doppler. Lack of compressibility, or direct identification of an endoluminal thrombus, were used as criteria for the diagnosis of thrombosis. Compression was performed in the transverse plane to avoid the probe sliding off the vessel wall on the longitudinal axis, potentially resulting in a false-negative

finding. Color Doppler imaging and spectral Doppler sonography (with an angle of insonation of 60˚ or less on the longitudinal axis of the vein) were used for the evaluation of subclavian veins, as established in the literature [17]. DVTs were considered as "catheter-related DVT" when occurring in the presence of an intravenous device. According to our local protocol, CUS was performed in each patient upon application of the c-pap helmet and repeated after 7 days of ventilation. In case of shorter duration of treatment with C-PAP, the examination was repeated when the helmet was removed. In addition, the examination was anticipated or possibly repeated in case of clinical suspicion of DVT. If DVT was diagnosed, no further CUS was performed.

## CPAP

Helmet CPAP was administered to patients who exhibited respiratory distress and a ratio of arterial partial pressure of oxygen to inspired fraction of oxygen (PaO2/FiO2) <200 mmHg during Venturi mask oxygen therapy or with a PaO2/Fio2 ratio between 200 and 300 mmHg, and clinical, radiological or laboratory evidence of severe disease. Subjects were included in the CPAP group if treated for at least 3 hours during hospitalization. Helmet CPAP was applied to patients and connected with a plastic tube to a source. The source delivered a mixture of oxygen and compressed air at flows to be adjusted according to a pre-defined table correlating these combined flows with a definite FiO2. Nurses regulated helmet CPAP and the resistance valve to a target SpO2 of at least 92%, progressing according to the following scheme: CPAP 5 cmH2O and 30% FiO2, CPAP 7 cmH2O and 40% FiO2, CPAP 10 cmH2O and 50% FiO2. The upper limit of performance was put at 15 cmH2O and 60% FiO2.

## Study outcome

The primary outcome of the study was the incidence of objectively diagnosed upper extremities deep venous thrombosis (UEDVT) including asymptomatic and symptomatic events. Lower extremities deep venous thrombosis (LEDVT), upper extremities superficial venous thrombosis (UESVT), lower extremities superficial venous thrombosis (LESVT), pulmonary embolism (PE) and mortality during hospitalization were also recorded.

## Statistical analysis

Continuous variables were expressed as median with interquartile range (IQR) or as mean plus or minus standard deviation (SD). Categorical variables were expressed as frequencies and percentages. Comparisons were performed between patients with UEDVT and those without. Categorical variables were compared using the chi-square test. Continuous variables were compared using Student's t test or Mann-Whitney test as appropriate. The normality of the data distribution was assessed by visually inspecting the histograms and Q-Q plots. The number of thrombotic events was expressed as a percentage (with 95% confidence intervals [CI] with Yates's continuity correction).

Univariate binomial logistic regressions were performed to evaluate the associations between patient characteristics and the occurrence of UEDVT. D-dimer was included both as a continuous variable and as a dichotomous variable based on the age-adjusted cut-off value (<500 μg/L up to 50 years of age, then age x 10 μg/L). Thrombophilia was not included in the association analysis as it was not routinely studied in all patients. Variables showing an association with the occurrence of UEDTV at univariate analysis (p <0.1) were included in stepwise backward multivariate logistic models. The significance level for variables removal was set at 0.05.

Two different models were generated. In model 1, patients with UEDVT were compared with subjects without UEDVT, regardless of the presence of LEDVT; in model 2, patients with UEDVT were compared with patients without DVT, therefore excluding patients with LEDVT from the control group, to evaluate their potential confounding effect on the associations found.

Furthermore, the association between patients characteristics and LEDVT, all cases of DVT and death was investigated as for UEDVT.

Results are reported as odds ratio (OR) along with their 95% confidence intervals (95% CI). Statistical significance was set at p<0.05. The STATA software for Mac, version 16 (Stata-Corp. 2019. Stata Statistical Software: Release 16. College Station, TX: StataCorp LLC.) was used for data processing.

### Ethical approval and informed consent

Due to the retrospective nature of the study, the local institutional Ethics Committee waived the need for informed consent and the ethical considerations of this research were conformed to the Declaration of Helsinki. The study was carried out and reported according to the Strengthening the Reporting of Observational Studies in Epidemiology (STROBE) guidelines for observational studies.

### Results

Between March/April and October/December 2020, 257 patients admitted to our Clinical Units because of COVID-19 met the inclusion criteria and were included in the study. Patient characteristics are shown in Table 1.

Mean age was 70.14 years (SD; 12.25) and 184 (70.0%) were male. Most of the included patients (97.28%) had at least one comorbidity: 141 (58.7%) patients had hypertension, 54 (22.5%) diabetes, 39 (15.2%) HF, 30 (11.8%) COPD and only 9 (3.5%) had a previous cerebro-vascular event. Moreover, 37 (14.5%) had a BMI > 30 Kg/m2, and 9 (3.5%) were affected by cancer. None of the patients had major thrombophilia while 22 (8.6%) had a minor congenital thrombophilia.

Helmet CPAP was applied to 164 (63.8%) patients with a median duration of ventilation of 9 days (IQR 5.5–14.5). The median duration of hospitalization was 16 days (IQR 9–27) and, at the end of data collection (December 4, 2020), 196 patients (76.0%) had been discharged, 4 (1.5%) transferred to another hospital, 53 had died (20.7%). and 3 (1.2%) were still hospitalized.

All patients (100.0%) were on anticoagulant thromboprophylaxis since the first day of hospitalization, in the majority of cases (96.1%) with enoxaparin 4000 UI (40 mg) SC OD and in a small number of cases (3.9%) with 2000 UI (20 mg) SC OD if creatinine clearance was under 30 mL/min. Venous CUS of the upper and lower extremities was performed on all enrolled patients between the 1st and the 30th day of hospitalization (median 7; IQR 4–9). Results are reported in Table 2.

UEDVT occurred in 28 patients (10.9%; 95% CI, 7.1–14.7) with 10 patients (35.7%) revealing concomitant LEDVT. Of these, 5 (50%) had a proximal LEDVT (involving the popliteal vein and/or proximal vessels), and 5 (50%) a distal DVT (infrapopliteal thrombosis without extension to proximal vessels). Isolated LEDVT was found in 17 patients (6.6%; 95% CI, 3.56–9.64) with an overall incidence of LEDVT of 10.5% (95% CI, 6.75–14.25). Isolated UESVT was diagnosed in 29 patients (11.3%; 95% CI, 7.4–15.2) while 16 patients (57.1%) were found to have concomitant UEDVT and UESVT. CTPA was performed in 32 patients (12.4%; 95% CI, 8.37–16.43) and PE was diagnosed in 9 (3.5%; 95% CI, 1.25–5.75).

**Table 1.**

|  | General Population (257) |
|---|---|
| Age, mean±SD | 70,14±12,25 |
| Male, n (%) | 184 (70,0%) |
| Days of hospitalization, mean±SD | 19,36±12,25 |
| Disease severity: |  |
| • PaO2/Fio2, mean±SD | 247,00±83,34 |
| • ARDS, n (%) | 120 (47,2%) |
| • Associated Pneumonia, n (%) | 208 (81,6%) |
| Patients requiring CPAP, n (%) | 164 (63,8%) |
| • Days of ventilation, mean±SD | 6,72±6,99 |
| Associated comorbidities: |  |
| • Heart Failure, n (%) | 39 (15,2%) |
| • COPD, n (%) | 30 (11,8%) |
| • Diabetes, n (%) | 54 (22,5%) |
| • Hypertension, n (%) | 141 (58,7%) |
| • History of Stroke, n (%) | 9 (3,5%) |
| • Dementia, n (%) | 24 (9,4%) |
| • Body mass index >30 kg/m2, n (%) | 37 (14.4%) |
| • Congenital thrombophilia, n (%) | 22 (8,63%) |
| • Cancer, n (%) | 9 (3,5%) |
| Blood test analysis: |  |
| • Creatinine, mean±SD | 1,04±0,62 |
| • Hemoglobin, mean±SD | 13,3±1,76 |
| • White cells count, mean±SD | 8,42±5,95 |
| • Platelet count, mean±SD | 236268,8±105361,7 |
| • Reattive C protein, mean±SD | 11,1±17,0 |
| • D-dimer, mean±SD | 3263,77±7506,65 |
| Deaths, n (%) | 53 (20,7%) |

Comparisons between patients with UEDVT and those without are reported in Table 3; 18 UEDVT patients (64.3%) were symptomatic for edema or pain of the upper extremities. Of all UEDVT, 8 (28.6%) were catheter-related and 9 (32.0%) were bilateral. The subclavian vein was involved in 3 patients (10.7%) with one patient showing bilateral thrombosis and another one catheter-related thrombosis. The axillary vein was involved in 25 patients (89.3%) with 9 (32.1%) patients showing bilateral thrombosis and 7 (15.9%) catheter-related thrombosis. No brachial or jugular thrombosis was reported. Patients with UEDVT did not differ from those without UEDVT in terms of age, sex, BMI or comorbidities. Patients with UEDVT were more likely to have a positive history of thrombophilia than patients without UEDVT (28.6% vs 6.17%; p<0.01).

UEDVT was more frequent in patients affected by more severe COVID19 as they showed significantly lower PaO2/FiO2 ratio (212.77±86.92 vs 251.35±82.06; p<0.05), increased incidence of associated pneumonia (96.4% vs 79.7%; p<0.05) and in a higher percentage met diagnostic criteria for ARDS (75.0% vs 43.8%; p<0.001). D-dimer levels at admission were higher in patients with UEDVT (8387.0±12049.26 vs 2614.0±6478.41; p<0.0001) with increased prevalence of subjects with levels above the reference range than non-UEDVT patients (89.3% vs 42.4%; p<0.001).

Patients with UEDVT more frequently required CPAP ventilation (92.9% vs 60.3%; p<0.0001), however there was no difference in the duration of ventilation between the two groups (days, 12.31±6.30 vs 10.18±5.93; p = n.s.).

**Table 2. Incidence and characteristics of UEDVT, LEDVT, UESVT and LESVT.**

|  | UEDVT 28 (10,9%) | LEDVT 27 (10.5%) | UESVT 29 (11.3%) | LESVT 8 (3.1%) |
|---|---|---|---|---|
| • Symptomatic | 18 (64.3%) | 16 (59.2%) | 13 (44.8%) | 3 (37.5%) |
| • Venous catheter related | 8 (28.6%) | 0 (0%) | 9 (31.0%) | 0 (0%) |
| • Bilateral | 9 (32.1%) | 0 (0%) | 4 (13.8%) | 0 (0%) |
| • Subclavian | 3 (10.7%) |  |  |  |
| • Axillary | 25 (89.3%) |  |  |  |
| • Brachial | 0 (0%) |  |  |  |
| • Jugular | 0 (0%) |  |  |  |
| • Concomitant LEDVT | 10 (35.7%) |  |  | 5 (62.5%) |
| • Concomitant UESVT | 16 (57.1%) |  |  | 3 (37.5%) |
| • Femoral |  | 11 (40.7%) |  |  |
| • Popliteal |  | 8 (29.6%) |  |  |
| • Distal lower limb deep veins |  | 8 (29.6%) |  |  |
| • Cephalic |  |  | 7 (24.1%) |  |
| • Basilic |  |  | 13 (44.8%) |  |
| • Distal upper arms superficial veins |  |  | 9 (31.0%) |  |
| • Great saphenous vein |  |  |  | 8 (100%) |
| • Small saphenous vein |  |  |  | 0 |

The number of deaths was significantly higher in COVID-19 patients who developed UEDVT than in those who did not (46.4% vs 17.5%, p <0.01).

The results of the univariate and multivariate analyses concerning model 1 and model 2 are reported in Table 4. The univariate analysis demonstrated the association between UEDVT and PaO2/FiO2 ratio (OR 0.99; 95% CI, 0.99–1.00); ARDS (OR 3.85; 95% CI, 1.57–9.42); helmet CPAP (OR 8.57; 95% CI, 1.99–37.00) and D-dimer above the age-adjusted range (OR 9.83; 95% CI, 3.59–26.89).

In the multivariate regression model, the occurrence of UEDVT was associated only with helmet CPAP application (OR 5.95; 95% IC, 1.33–26.58) and D-dimer above the age-adjusted range (OR 8.20; 95% IC, 2.94–22.89). The results obtained with model 2 were similar to those of model 1 (Table 4).

The diagnosis of LEDVT was made in 27 patients (10.5%. 95% CI, 6.75–14.25) and of those, 16 (59.2%) were symptomatic. Proximal LEDVT was found in 19 patients (70.4%) while in 8 cases (29.6%) only distal veins were involved. (Table 2 and S1 Table in S1 Appendix).

Overall, DVT was found in 45 patients (17.5%, 95% CI, 12.85–22.15) and was more frequent in patients with more severe respiratory failure, as they showed lower PaO2/FiO2 ratio (216.88±77.56 vs 253.60±83.29; p<0.01) and more frequently ARDS (66.7% vs 42.4%; p<0.01), regardless of the presence of pneumonia. Accordingly, DVT patients more frequently required helmet CPAP ventilation (82.2% vs 59.9%; p<0.01). Patients with DVT more frequently had a positive D-dimer (above the age-adjusted reference range, 86.7% vs 26.9%; p<0.0001) and an overall higher mean D-dimer (7671.65±10102.49 vs 2255.6±6394.91; p<0.0001) than patients without DVT. Multivariate analysis identified helmet CPAP (OR 5.75; 95% CI, 1.27–26.07) and D-dimer above the age-adjusted value (10.40; 95% CI, 3.69–29.30) as variables associated with DVT. (S2 Table in S1 Appendix).

The number of deaths was 53 (20.6%; 95% CI, 15.66–25.54) and PE was considered the most likely cause in 5 patients (S3 Table in S1 Appendix). At the multivariate analysis, death was associated with age >65 years (OR 8.33; 95% CI, 2.18–31.8), helmet CPAP application (OR 3.32; 95% CI, 1.15–9.58), dementia (OR 20.16; 95% CI, 5.66–71.75) and occurrence of

**Table 3. Comparison between patients with UEDVT and patients without.**

| | UEDVT (28) | No-UEDVT (229) | p |
|---|---|---|---|
| Age, mean±SD | 73.0±9.76 | 69.79±12.49 | n.s. |
| Male, n (%) | 21 (75.0%) | 163 (71.2%) | n.s. |
| Days of hospitalization, mean±SD | 22.28±14.34 | 19.0±11.96 | n.s. |
| Disease severity: | | | |
| • PaO2/Fio2, mean±SD | 212.78±86.92 | 251.35±82.06 | <0.05 |
| • ARDS, n (%) | 21 (75.0%) | 99 (43.8%) | <0.05 |
| • Associated Pneumonia, n (%) | 27 (96.4%) | 181 (79.7%) | <0.05 |
| Patients requiring CPAP, n (%) | 26 (92.9%) | 138 (60.3%) | <0.0001 |
| • Days of ventilation, mean±SD | 12.31±6.30 | 10.18±5.93 | n.s. |
| Associated comorbidities: | | | |
| • Heart Failure, n (%) | 5 (17.9%) | 34 (14.9%) | n.s. |
| • COPD, n (%) | 1 (3.6%) | 29 (12.8%) | n.s. |
| • Diabetes, n (%) | 3 (11.1%) | 51 (23.9%) | n.s. |
| • Hypertension, n (%) | 15 (55.6%) | 126 (59.1%) | n.s. |
| • History of Stroke, n (%) | 0 (0%) | 9 (3.9%) | n.s. |
| • Dementia, n (%) | 2 (7,1%) | 22 (9.6%) | n.s. |
| • Body mass index >30 kg/m2, n (%) | 4 (14.3%) | 33 (14.5%) | n.s. |
| • Cancer, n (%) | 0 (0%) | 9 (3.9%) | n.s. |
| • Thrombophilia, n (%) | 8 (28.6%) | 14 (6.1%) | <0.001 |
| Blood test analysis: | | | |
| • Creatinine, mean±SD | 1.06±0.58 | 0.99±0.30 | n.s |
| • Hemoglobin, mean±SD | 13.69±1.81 | 13.52±1.61 | n.s |
| • White cells count, mean±SD | 10.62±4.83 | 8.62±7.26 | n.s. |
| • Platelet count, mean±SD | 267185.2±92443.35 | 232022.1±98393.05 | n.s |
| • Reattive C protein, mean±SD | 13.81±6.09 | 12.77±22.25 | n.s |
| • D-dimer, mean±SD | 8387.0±12049.26 | 2614.0±6478.41 | <0.001 |
| • D-dimer > age adjusted range, n (%) | 23 (82.1%) | 73 (31.9%) | <0.001 |
| Deaths, n (%) | 13 (46.4%) | 40 (17.5%) | <0.01 |

**Table 4. Univariate and multivariate analyses of the association between UEDVT and potential risk factors.**

| Univariate analysis | Model 1 | | | Model 2 | | |
|---|---|---|---|---|---|---|
| Variable | OR | 95% CI | p | OR | 95% CI | P |
| PaO2/FiO2 | 0.99 | 0.99–1.00 | 0.022 | 0.99 | 0.99–1.00 | 0.018 |
| ARDS | 3.85 | 1.57–9.42 | 0.003 | 3.97 | 1.61–9.74 | 0.003 |
| Pneumonia | 6.86 | 0.91–51.83 | 0.062 | 6.95 | 0.92–52.61 | 0.060 |
| CPAP | 8.57 | 1.99–37.00 | 0.004 | 8.70 | 2.01–37.62 | 0.004 |
| Days of ventilation | 1.10 | 1.04–1.16 | 0.001 | 1.10 | 1.04–1.16 | 0.001 |
| D-dimer (age-adjusted) | 9.83 | 3.59–26.89 | 0.001 | 12.51 | 4.54–34.47 | 0.000 |
| **Multivariate analysis** | Model 1 | | | Model 2 | | |
| Variable | OR | 95% CI | p | OR | 95% CI | P |
| CPAP | 5.95 | 1.33–26.58 | 0.019 | 5.75 | 1.27–26.07 | 0.023 |
| D-dimer (age-adjusted) | 8.20 | 2.94–22.89 | 0.000 | 10.40 | 3.69–29.30 | 0.000 |

Model 1 represents the comparison between patients with UEDVT and patients without UEDVT; model 2 represents the comparison between patients with UEDVT and patients without DVT.

UEDVT (OR 3.71; 95% CI, 1.41–9.78). The diagnosis of LEDVT or DVT in general was not associated with increased mortality.

UESVT was found in 44 patients (17.1%; 95% CI, 12.5–21.7), while LESVT in 8 (3.1%; 95% IC, 0.98–5.22). Detailed description of UESVT and LESVT is reported in Table 2.

## Discussion

Our study, that included a number of patients affected by moderate-severe COVID-19, provides some relevant findings. First, we reported a high incidence of UEDVT despite the recommended antithrombotic prophylaxis. Second, patients under helmet CPAP showed a 6 times increased risk of UEDVT compared with those who did not require ventilation. Third, COVID-19 patients who developed an UEDVT showed an increased risk of death compared to those who did not.

Previous studies have reported a high risk of DVT in non-ICU COVID-19 patients despite the recommended thromboprophylaxis, with an incidence estimated at 2.4% when only proximal veins are included [18], up to 14.7% when also distal veins are included [19]. In the majority of these studies, a systematic screening of the deep venous system of the lower limbs was performed while the exploration of the upper extremities was generally avoided or only limited to symptomatic patients (e.g. those complaining of upper limb pain or swelling). For instance, in a retrospective analysis on 188 COVID-19 patients who had undergone upper or lower extremity Doppler ultrasound, Chang and coll. reported an incidence of DVT of 31%. Out of a total of 58 DVTs, only 4 were located in the upper extremity venous system, with an incidence of 2.1% [20]. However, not all patients included in the study had been subjected to both upper and lower limbs ultrasound and this may have caused an underestimation of the number of UEDVT. In our study the incidence of LEDVT (10.5%) was similar to that reported by them or other groups in the same type of patients [18, 19], conversely we reported a significantly higher incidence of UEDVT (10.9%) with one-third of affected patients asymptomatic for upper limb edema or pain. It is likely that, without a systematic examination of the upper extremities, a not negligible number of DVTs probably not have been identified, leading to an underestimation of their incidence. In the general population, UEDVTs usually occur in the presence of a provoking condition, such as the presence of a central venous catheter or other intravascular devices [21–28]. In our cohort only 8 UEDVT (28.6%) were catheter-related and the presence of venous catheters was not associated with the onset of UEDVT. In a non-COVID-19 setting, non-catheter-related UEDVTs generally happen in the presence of risk factors such as cancer, recent surgery, pregnancy, oral contraceptive use, or immobilization of the arm [29, 30]; however, none of the above was recorded or correlated with thrombosis in our patients.

In our study, patients with UEDVT did not differ from those without UEDVT in terms of age, male sex, and comorbidities. Of note, obesity, which has been associated with a worse prognosis in COVID-19 patients [4] and with an increased incidence of VTE [31, 32], was not significantly more frequent in patients with UEDVT. Moreover, white cell count, platelet count, hemoglobin and CRP levels were similar between patients with and without DVT and this evidence suggests that the level of inflammation does not correlate with the incidence of UEDVT.

We found a significantly higher level of D-dimer in patients with UEDVT, and a higher prevalence of patients with D-dimer levels above the normal range. Importantly, evidence of abnormal D-dimer levels was associated with the diagnosis of UEDVT, but the proposed cut-off of 1500 for DVT in COVID-19 patients [33] was not predictive of UEDVT. Furthermore,

the increase in D-dimer must be considered a consequence of the presence of thrombosis and not a risk factor for its development.

Severe pneumonia and ARDS have been previously associated with DVT in COVID-19 infection [34]. In this study, UEDVT was more frequent in subjects with severe pulmonary infection but its incidence did not correlate with pneumonia or ARDS in the regression model. On the contrary, ventilation with helmet CPAP was associated with six times the risk of UEDVT. The application of mechanical ventilation, especially with high positive end-expiratory pressure (PEEP), may cause a reduction in venous return and consequent venous stasis [35]. Prolonged mechanical ventilation has been already associated with high incidence of DVT despite thromboprophylaxis in critically ill patients admitted to ICU [36]. Ibrahim and coll. reported an incidence of 4.5% of UEDVT in a cohort of 110 patients under mechanical ventilation, which is lower than the one we found in our study. One possible explanation for this discrepancy may be found in the different ventilation systems. All of our patients were treated with helmet CPAP which is generally preferred over a face mask because it avoids skin lesions and pain, reduces discomfort, and improves patient tolerance. Also, helmet CPAP seems to reduce SARS-CoV-2 aerosolization and virus transmission compared to facial mask CPAP [37]. The helmet is held in place by two straps placed under the armpits and it can be hypothesized that this anchoring system causes mechanical compression of the axillary veins and promotes the activation of the thrombotic process. Interestingly, most UEDVTs were isolated axillary venous thrombosis which is an unusual presentation of UEDVT, of which a high percentage were bilateral and occurred in the absence of central venous catheters. These evidences suggest that the mechanical and hemodynamic effect of the compression of the axillary veins by the CPAP helmet anchorage system appears to play a central role and to be the main provoking factor for these high numbers of UEDVTs. Although in the general population, the risk of PE in patients with UEDVT is considered to be lower than that with LEDVT [38–42] the high incidence of UEDVT in COVID-19 patients may explain, at least in part, the paucity of DVT observed in COVID-19 patients with PE [34]. Furthermore, it has been demonstrated that several mechanisms may contribute to a hypercoagulable state in COVID-19 infection [43] and that affected patients show a coagulation activation pattern different from that observed in critically ill patients [14], therefore, in this setting, the real risk of embolization from UEDVT is still unclear.

VTE has been implicated in a not negligible proportion of deaths due to coronavirus 2019; for instance, in a study on 80 death for COVID-19, PE was found in 17 cases (21%) and was considered the cause of death in 8 (9.9%) [44]. The effect of DVT on COVID-19 prognosis has been already reported [45] with an increased need for admission to the ICU and higher mortality. In our study, we found an overall mortality of 20.7% and demonstrated that UEDVT was associated with a fourfold increased risk of death. As post-mortem examination of patients who died for COVID-19 is generally avoided, also due to the persistent risk of contagion, none of our patients underwent an autopsy. However, we found 10 PE and the clinical suspicion of death for PE was very high in at least 5 patients (9.4%) with 4 of them showing UEDVT but not LEDVT.

To our knowledge, this is the first study that demonstrates that the occurrence of UEDVT in non-ICU COVID-19 patients may worsen the course of the disease. Furthermore, mortality was not associated with LEDVT or DVT in general, and this evidence increases the need for greater attention to the veins of the upper arms in COVID-19 patients and in particular in those undergoing helmet CPAP ventilation.

Unfortunately, we cannot compare the incidence of UEDVT in our cohort of COVID-19 patients with that by a hypothetical group of patients treated with facemask-CPAP, therefore our findings apply only to patients with helmet-CPAP and this is one of the limitations of this study. Another potential limitation of the study was that CUS was performed once during the

hospitalization and not at the same time-point for all patients. However, this drawback could have caused at most an underestimation of the events. Additionally, we noted an increased prevalence of minor thrombophilia in patients with UEDVT, but this evidence is limited by the fact that our patients were not routinely screened for thrombophilia, indeed only previous diagnoses were recorded. Finally, the main limitation of our study is its retrospective nature; however, according to our local COVID-19 protocol management, all the patients admitted to our unit are systematically screened for UEDVT and LEDVT, all comorbidities are carefully recorded and they undergo the same battery of laboratory tests. Thus the number of missing values in this study is very low.

## Conclusions

The evidence of the increased incidence of UEDVT in COVID-19 patients under helmet CPAP raises questions about the right antithrombotic prophylaxis to be given to these patients. Few months ago, the American Society of Hematology 2021 guidelines on the use of anticoagulation for thromboprophylaxis in patients with COVID-19 have been published [46] and, despite the evidence of a higher rate of VTE in acutely ill or hospitalized COVID-19 patients, prophylactic-intensity anticoagulation over intermediate-intensity or therapeutic-intensity anticoagulation is still recommended in subjects who do not have confirmed or suspected VTE. Nonetheless, the use of an intermediate dose of LMWH (e.g., enoxaparin 4000 IU subcutaneously every 12 h) has been suggested by panels of experts on an individual basis in patients with multiple risk factors for VTE [47]. Administration of a prophylactic intermediate-dose anticoagulant to COVID-19 ICU patients has been reported as not superior to a standard dose for the composite outcome of venous or arterial thrombosis, need for extracorporeal membrane oxygenation, or 30 days mortality. In addition, patients treated with higher doses showed an increased risk of major bleeding and heparin-induced thrombocytopenia compared with those on standard dose [48]. Although this evidence is meaningful, most of the patients included in the study were in an extremely advanced and probably irreversible stage of the disease. Smadja and coll. have reported that among COVID-19 patients hospitalized in medical wards, intermediate-dose anticoagulant compared with standard-dose anticoagulant did not result in a significant difference in in-hospital mortality, however only the 3,3% of the patients included in their retrospective analysis required CPAP ventilation and less than 20% had severe disease at lung CT-scan [49]. In contrast, 3 randomized controlled trials reported improved clinical outcomes in patients with moderate COVID-19 disease empirically prescribed with treatment-dose anticoagulation [50–52]. So that, based on these data, the NICE (National Institute for Health and Care Excellence) guidelines on the management of COVID-19 disease has been recently updated by introducing the recommendation to consider prescribing a therapeutic dose of LMWH for young people and adults with COVID-19 who require low-flow oxygen and who do not have an increased risk of bleeding [53].

Therefore, the hypothesis that an intermediate dose of heparin may be effective in COVID-19 patients at high risk of DVT who are not admitted to the ICU and with an early stage of the disease may still be valid.

In this study, COVID-19 patients requiring CPAP showed a high risk of UEDVT; nevertheless, further investigation is required to evaluate the risk/benefit of an intermediate dose of heparin in these patients.

## Supporting information

**S1 Appendix.**
(DOCX)

**S1 Database.**
(XLS)

## Author Contributions

**Conceptualization:** Nicola Mumoli, Francesco Dentali, Igor Giarretta.

**Data curation:** Nicola Mumoli, Francesco Dentali, Francesca Zuretti, Marco Cei, Igor Giarretta.

**Formal analysis:** Nicola Mumoli, Francesco Dentali, Igor Giarretta.

**Investigation:** Nicola Mumoli, Francesco Dentali, Giulia Conte, Alessandra Colombo, Riccardo Capra, Cesare Porta, Giuseppe Rotiroti, Marco Cei, Isabella Evangelista.

**Methodology:** Nicola Mumoli, Francesco Dentali, Francesca Zuretti, Marco Cei, Flavio Tangianu, Josè Vitale, Antonino Mazzone, Igor Giarretta.

**Supervision:** Nicola Mumoli, Francesco Dentali, Marco Cei, Flavio Tangianu, Antonino Mazzone, Igor Giarretta.

**Validation:** Nicola Mumoli, Francesco Dentali, Josè Vitale, Antonino Mazzone.

**Visualization:** Francesco Dentali, Flavio Tangianu.

**Writing – original draft:** Nicola Mumoli, Francesco Dentali, Igor Giarretta.

**Writing – review & editing:** Nicola Mumoli, Francesco Dentali, Francesca Zuretti, Marco Cei, Josè Vitale, Igor Giarretta.

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
