## [Decision Letter · Decision Letter 0]

15 Dec 2021

PONE-D-21-22254Upper extremity deep vein thrombosis in COVID-19: incidence and correlated risk factors in a cohort of non-ICU patientsPLOS ONE

Dear Dr. Mumoli,

Thank you for submitting your manuscript to PLOS ONE. After careful consideration, we feel that it has merit but does not fully meet PLOS ONE’s publication criteria as it currently stands. Therefore, we invite you to submit a revised version of the manuscript that addresses the points raised during the review process.

 Please see reviewer comments below. Minor modifications are needed. 

We look forward to receiving your revised manuscript.

Kind regards,

Amit Bahl

Academic Editor

PLOS ONE

Journal Requirements:

Reviewers' comments:

Reviewer's Responses to Questions

**Comments to the Author**

1. Is the manuscript technically sound, and do the data support the conclusions?

Reviewer #1: Yes

Reviewer #2: Yes

2. Has the statistical analysis been performed appropriately and rigorously? 

Reviewer #1: Yes

Reviewer #2: Yes

3. Have the authors made all data underlying the findings in their manuscript fully available?

Reviewer #1: Yes

Reviewer #2: Yes

4. Is the manuscript presented in an intelligible fashion and written in standard English?

Reviewer #1: Yes

Reviewer #2: Yes

5. Review Comments to the Author

Reviewer #1: Comments:

- Venous CUS of the upper and lower extremities was performed on all enrolled patients between the 1st and the 30th day of hospitalization (median 7; IQR 4-9). This procedure of screening for DVT needs to be better described. When exactly was it done? Why was it done at that time? What exactly does it say in “the local COVID-19 patient management protocol”?

- The most striking finding of this work is the presence of a majority of isolated axillary vein thrombosis (which is an unusual presentation of UEDVT), of whom a high proportion are bilateral, and occur in the absence of central venous catheters. So the mechanical/haemodynamic effect of compression of the axillary veins by the anchoring system of the helmet CPAP system is likely to play a central role and be the main provoking factor for these unusual UEDVTs. This should be highlighted more clearly throughout the manuscript.

- One important limitation is that the results and findings apply only to patients with helmet CPAP. To be stated more clearly.

Reviewer #2: Interesting manuscript, well supported with clear discussion. The paper is well written and formally

correct and it is clear its clinical relevance, and what this article should add to the body of

knowledge on this topic.

Minor Comments:

- In this study most DVTs are UEDVT and in particular axillary venous thrombosis, there are several

possible explanations for this finding. Is the UEDVT due to the increase in venous pressure induced

by the application of Helmet CPAP or is it strictly related to the mechanical compression of caused

by the anchor strips? Authors should discuss this.

- In the paragraph Conclusion the authors state that the hypothesis that an intermediate dose of

heparin may be effective in COVID-19 patients at high risk of DVT not hospitalized in intensive care

and with an early stage of the disease may still be valid. However, a number of recent publications

seem to disagree with this. For example, a recent article by Samdja et al. reported that among

COVID-19 patients admitted to medical wards, intermediate-dose prophylactic anticoagulation

versus standard-dose prophylactic anticoagulation did not result in a significant difference in

hospital mortality. This document needs to be cited and discussed.

- In this study, obesity, which has been associated with a worse prognosis in COVID-19 infection, is

not associated with UEDVT, it can be hypothesized that a larger arm diameter may be "protective"

on axillary vein compression caused by the helmet straps?

6. PLOS authors have the option to publish the peer review history of their article (what does this mean?). If published, this will include your full peer review and any attached files.

Reviewer #1: No

Reviewer #2: **Yes: **di micco p

---

## [Author Response · Author response to Decision Letter 0]

21 Dec 2021

Dicember 20th 2021

Dr. Amit Bahl

Academic Editor

PLOS ONE 

Re: PONE-D-21-22254

Dear Dr. Bahl,

I am writing in response to your email of December 14th 2021 regarding the above-mentioned manuscript.

We are grateful that Reviewers for their positive comments to our paper and for defining it as interesting and well supported with clear data. We are also grateful to the Reviewers for their constructive criticisms. Therefore, we have edited the manuscript according to their suggestions.

Below, our answers to all the Reviewers’ comments are listed point-by-point, along with a description of the changes that were made to the previous version of the manuscript.

We hope that this manuscript in now acceptable for publication in the Plos One.

Sincerely,

Nicola Mumoli

 

Reviewer: 1

1. Venous CUS of the upper and lower extremities was performed on all enrolled patients between the 1st and the 30th day of hospitalization (median 7; IQR 4-9). This procedure of screening for DVT needs to be better described. When exactly was it done? Why was it done at that time? What exactly does it say in “the local COVID-19 patient management protocol”?

According to our local protocol, CUS was performed in each patient upon application of the c-pap helmet and repeated after 7 days of ventilation. In case of shorter duration of treatment with C-PAP, the examination was repeated when the helmet was removed. In addition, the examination was anticipated or possibly repeated in case of clinical suspicion of DVT. If DVT was diagnosed, no further CUS was performed. Furthermore, the c-pap helmet was not always applied at hospitalization but in a percentage of cases it was prescribed for evidence of respiratory distress.

This approach explains the large interval on the day of CUS administration, however the median exam day was 7th with 75% of the population ranging from 4th to 9th day.

2. The most striking finding of this work is the presence of a majority of isolated axillary vein thrombosis (which is an unusual presentation of UEDVT), of whom a high proportion are bilateral, and occur in the absence of central venous catheters. So the mechanical/haemodynamic effect of compression of the axillary veins by the anchoring system of the helmet CPAP system is likely to play a central role and be the main provoking factor for these unusual UEDVTs. This should be highlighted more clearly throughout the manuscript. 

We thank the reviewer for this comment and we also agree with his statement. Accordingly we added to the discussion the following sentence:…Interestingly, most UEDVTs were isolated axillary venous thrombosis which is an unusual presentation of UEDVT, of which a high percentage were bilateral and occurred in the absence of central venous catheters. These evidences suggest that the mechanical and hemodynamic effect of the compression of the axillary veins by the CPAP helmet anchorage system appears to play a central role and to be the main provoking factor for these high numbers of UEDVTs…

We also removed the following sentence from the text: “According to this hypothesis we also found a higher incidence of UESVT compared to LESVT” which seems to be pleonastic after the suggested changes. 

3. One important limitation is that the results and findings apply only to patients with helmet CPAP. To be stated more clearly.

We agree with this reviewer 1 comment, however we must point-out that, as reported in the text, the majority of COVID-19 patients are ventilated with helmet-CPAP for a number of reasons so our findings appear to be clinical relevant. We integrated the text as follows:

“Unfortunately, we cannot compare the incidence of UEDVT in our cohort of COVID-19 patients with that by a hypothetical group of patients treated with facemask-CPAP, therefore our findings apply only to patients with helmet-CPAP and this is one of the limitations of this study”.

Reviewer: 2

1. In this study most DVTs are UEDVT and in particular axillary venous thrombosis, there are several possible explanations for this finding. Is the UEDVT due to the increase in venous pressure induced by the application of Helmet CPAP or is it strictly related to the mechanical compression of caused by the anchor strips? Authors should discuss this.

Se answer 2 to reviewer 1. 

2. In the paragraph Conclusion the authors state that the hypothesis that an intermediate dose of heparin may be effective in COVID-19 patients at high risk of DVT not hospitalized in intensive care and with an early stage of the disease may still be valid. However, a number of recent publications seem to disagree with this. For example, a recent article by Samdja et al. reported that among COVID-19 patients admitted to medical wards, intermediate-dose prophylactic anticoagulation versus standard-dose prophylactic anticoagulation did not result in a significant difference in hospital mortality. This document needs to be cited and discussed.

We are grateful to the reviewer for this comment. The text has been revised with the addition of the following sentence and the inclusion of the suggested article in the references: “Smadja and coll. have reported that among COVID-19 patients hospitalized in medical wards, intermediate-dose anticoagulant compared with standard-dose anticoagulant did not result in a significant difference in in-hospital mortality, however only the 3,3% of the patients included in their retrospective analysis required CPAP ventilation and less than 20% had severe disease at lung CT-scan”. 

3. In this study, obesity, which has been associated with a worse prognosis in COVID-19 infection, is not associated with UEDVT, it can be hypothesized that a larger arm diameter may be "protective" on axillary vein compression caused by the helmet straps?

This is a very interesting hypothesis raised by the Reviewer 2, which seems to be reasonable but hardly to demonstrate. In addition, only 14% of the patients included in our study had a BMI greater than 30 therefore, we do not consider it appropriate to discuss this hypothesis in the main text.

---

## [Editor Report · Decision Letter 1]

28 Dec 2021

Upper extremity deep vein thrombosis in COVID-19: incidence and correlated risk factors in a cohort of non-ICU patients

PONE-D-21-22254R1

Dear Dr. Mumoli,

We’re pleased to inform you that your manuscript has been judged scientifically suitable for publication and will be formally accepted for publication once it meets all outstanding technical requirements.

Kind regards,

Amit Bahl

Academic Editor

PLOS ONE
---

## [Editor Report · Acceptance letter]

3 Jan 2022

PONE-D-21-22254R1 

Upper extremity deep vein thrombosis in COVID-19: incidence and correlated risk factors in a cohort of non-ICU patients 

Dear Dr. Mumoli:

I'm pleased to inform you that your manuscript has been deemed suitable for publication in PLOS ONE. Congratulations! Your manuscript is now with our production department. 

Kind regards, 

on behalf of

Dr. Amit Bahl 

Academic Editor

PLOS ONE